# Difference in Incontinence Pad Use between Patients after Radical Prostatectomy and Cancer-Free Population with Subgroup Analysis for Open vs. Minimally Invasive Radical Prostatectomy: A Descriptive Analysis of Insurance Claims-Based Data

**DOI:** 10.3390/ijerph18136891

**Published:** 2021-06-27

**Authors:** Dong-Ho Mun, Lin Yang, Shahrokh F. Shariat, Sylvia Reitter-Pfoertner, Gerald Gredinger, Thomas Waldhoer

**Affiliations:** 1Department of Urology, Medical University of Vienna, 1090 Vienna, Austria; dong-ho.mun@meduniwien.ac.at (D.-H.M.); sfshariat@gmail.com (S.F.S.); 2Department of Cancer Epidemiology and Prevention Research, Alberta Health Services, Calgary, AB T2S 3C3, Canada; linyang33@gmail.com; 3Departments of Oncology and Community Health Sciences, University of Calgary, Calgary, AB T2N 1N4, Canada; 4Institute for Urology and Reproductive Health, I.M. Sechenov First Moscow State Medical University, 119435 Moscow, Russia; 5Department of Urology, Weill Cornell Medical College, New York, NY 10065, USA; 6Department of Urology, University of Texas Southwestern, Dallas, TX 75390, USA; 7Karl Landsteiner Institute of Urology and Andrology, 3100 St. Poelten, Austria; 8Department of Urology, Second Faculty of Medicine, Charles University, 150 06 Prague, Czech Republic; 9Department of Special Surgery, Division of Urology, Jordan University Hospital, The University of Jordan, Amman 2V89+CR, Jordan; 10European Association of Urology Research Foundation, NL-6803 AA Arnhem, The Netherlands; 11Competence Center Integrated Care, c/o Austrian Health Insurance Fund, 1100 Vienna, Austria; sylvia.reitter-pfoertner@oegk.at (S.R.-P.); Gerald.Gredinger@goeg.at (G.G.); 12Center for Public Health, Department of Epidemiology, Medical University of Vienna, 1090 Vienna, Austria

**Keywords:** insurance data, laparoscopic radical prostatectomy, open radical prostatectomy, robotic radical prostatectomy, urinary incontinence

## Abstract

Purpose: to quantify and compare pre- and post-surgical incontinence pad use between men treated with radical prostatectomy (RP) for prostate cancer (PCa) and cancer-free controls, using population-based Austrian insurance claims data. Methods: Men who underwent RP for treating PCa between 2013–2015 were identified. Cancer-free men ≥45 years with and without benign prostate hyperplasia (BPH) were used as controls. Longitudinal data on ICD-diagnoses, type of surgery, prescribed incontinence pads, and hospitals’ surgery volumes were aggregated between 2011–2018 to capture pre- and up to three years post-RP follow-up. Monthly rates of pad use were calculated and compared between RP types and cancer-free controls. Results: A total of 6248 RP patients, 7158 cancer-free men with BPH, and 50,257 cancer-free men without BPH were analyzed. Comparing to pre-RP (0.03, 95%CI: 0.02–0.05), RP resulted in significantly higher rates of prescribed pads (at 3 months: 12.61, 95%CI: 11.59–13.65; 12 months: 6.71, 95%CI: 6.10–7.34; 36 months: 4.91, 95%CI: 3.76–4.62). These rates were also higher than those for cancer free controls (with BPH:0.06, 95%CI: 0.04–0.09; without BPH:0.12, 95%CI: 0.10–0.14). The rate of prescribed pads after surgery continued to decline over time and remained higher among men who underwent minimally invasive RP compared to those who underwent an open procedure. Conclusion: Despite progress in surgical techniques, post-RP incontinence remains a prevalent adverse event. The rate of pad usage steadily improved over the first three years post RP. The rate of patients with incontinence needing pads was higher among those who were treated minimally invasive compared to open approach.

## 1. Introduction

Radical prostatectomy (RP), the standard surgical treatment for non-metastatic prostate cancer (PCa), yields satisfactory durable oncological outcomes [1,2]. For patients, functional outcomes are essential in their treatment satisfaction and quality of life (QoL). Urinary incontinence (UI) is a common side-effect of RP [3,4], ranging between 2% and 60% across studies [5]. While many patients eventually reach satisfactory urinary continence during recovery [6], a non-negligible proportion experience lasting postoperative UI that significantly affects their QoL [7]. Over time, technique advancements including minimally invasive surgery (MIS) (laparoscopic, robotic etc.) have gained in popularity with the promise to reduce functional adverse events of RP [8,9,10].

The real-world impact of UI after RP is an important endpoint for quality, cost, and patient satisfaction as part of value-based healthcare delivery, but difficult to measure. A major challenge is the lack of a clear definition and standardized metrics. Patient-reported outcome measures (PROMs) are common methods that capture patients’ perception of symptoms. However, there is no consensus on such measures [11] and a wide discrepancy exists in the perception of the same adverse event [12,13]. Therefore, incontinence pad use has been considered as a reproducible and reliable objective assessment of the presence and severity of urinary leakage that also reflects resource utilization [3,13].

Austria is one of the few countries where national-wide healthcare data are available, supported by the public and equal access healthcare system with compulsory universal health coverage [14]. Using population-based longitudinal data retrieved from insurance claims in Austria, our aims were to (1) quantify the pre- and post-RP incontinence pad use among PCa men and compare to cancer-free controls over 36 months, (2) compare the pre- and post-RP incontinence pad use between MIS vs. open RP (ORP) within PCa men.

## 2. Methods

### 2.1. Data

Data were retrieved from Austrian insurance claims between 2011 and 2018 through the Competence Center Integrated Care—of the Austrian health insurance fund [15]. This system covers 99.9% Austrians through 29 insurance schemes [16]. The aggregated data encompass all insurance claims within Austria and provide information on hospital level RP volume, and patient level hospital admissions including ICD-diagnosis, length of stay, type of in-patient medical service and prescribed medical aids (e.g., incontinence pads). Ethical approval was obtained (1867/2018).

### 2.2. Measures of Incontinence Pads

Longitudinal data on in-patient stays and prescribed medical aids were retrieved for each participant. Pad prescription was defined as medical aid coded as “saugende Inkontinenz” (absorbing incontinence aid). Date of prescription and number of prescribed pads was retrieved for each claim.

### 2.3. Selection of Patients

To obtain a sizable sample with considerable post-RP follow-up (36 months), men with PCa who underwent RP between 2013–2015 were selected. During the same time-period, two groups of cancer-free men with non-operated benign prostate hyperplasia (BPH) and without BPH were additionally selected as control groups to facilitate comparisons.

In accordance with predefined criteria (for detailed methods, please see Appendix A), participants were categorized into three groups: (1) men with PCa and underwent RP; (2) cancer-free men with non-operated BPH; (3) cancer-free men without BPH. RP was further classified according to the technique used into “open” vs. “minimally invasive” using the reimbursement codes “RPE OPEN” and “RPE LAP”, where laparoscopic procedure includes both robotic-assisted and standard laparoscopic prostatectomy.

### 2.4. Analysis

The analysis was done by SAS version 9.4 (SAS Institute Inc., Cary, NC, USA). Sample sizes and age for PCa patients and cancer free controls were summarized using counts and medians with interquartile ranges. Patient use of incontinence pad was assessed by dichotomized pad prescription status (yes/no), total number (cumulative sum) of pad prescriptions, total number of prescribed pads and date of prescriptions, and the daily rate (cumulative sum per day) of prescribed pads. Monthly rate of prescribed pads was calculated by multiplying the daily rate by 30.42 (average days in one month).

To describe the short- and long-term patterns of pad use among men who underwent RP for PCa, four time points were defined: “before RP”, “three months (90 days)”, “12 months (365 days)” and “36 months (1095 days)” after RP. For controls, rates of prescribed pad were calculated based on the period 2011–2018. For each time point, rates of pad use and corresponding 95% confidence intervals (95% CIs) were calculated for participants in each of the three groups (RP [ORP, MIS, combined], cancer-free men with and without BPH).

Life table analyses were done to plot the pad prescription status using proportion of PCa men who were prescribed any incontinence pads from the surgery day through 36-month follow-up for MIS and ORP, respectively. To describe the temporal pattern of pad use in greater detail, the monthly rate of prescribed pads was calculated restricting to those who ever received a pad prescription during the study phase. Firstly, daily rate of prescribed pad was calculated and smoothed by a generalized additive model (proc GAM in SAS). Based on the smoothed cumulative sum, the pad rate was estimated by the difference between two time points divided by the length of the time interval for stable pad rates. Secondly, monthly rates of prescribed pad were calculated and presented by age group (45–<60, 60–<70 and ≥70 years). The age effect was investigated using Poisson regressions adjusting for the pre-RP monthly rate of prescribe pads for MIS and ORP, respectively. Third, the total number of prescribed pads was calculated for each patient and categorized into 1–499, 500–599, 1000–1499, 1500–1999, 2000–2499, 2500–2999, and 3000 pads and more over 36 months. The distribution of these categories was presented using percentages for MIS and ORP, respectively. Finally, operation volumes (number of ORP and MIS RP operations) were calculated for each hospital and collapsed into <250 and ≥250 RP operations within 2013–2015 to prevent the identifiability of corresponding hospitals. The effect of operation volume per hospital on pads use was examined using the Spearman’s correlation coefficient.

## 3. Results

Sample characteristics and incontinence pad prescription are detailed in Table 1. A total of 6248 PCa patients (3262 ORP [52.2%, 64.6 years], 2986 MIS [47.8%, 64.6 years]) underwent RP between 2013–2015. Before surgery, 15 (0.5%) ORP and 36 (1.2%) MIS RP patients received pad prescription. These numbers rose to 175 (5.4%) and 737 (24.7%) immediately postoperatively, respectively (Figure 1). For the two cancer-free control groups, 54 out of 7157 men with non-operated BPH (70.8 years) and 504 out of 50,257 without BPH (64.3 years) received any pad prescription.

Total number of pad prescriptions, total number and rate of prescribed pads are presented restricting to those who received at least one prescription (Table 1).

### 3.1. Effect of Type of RP

With respect to the monthly rate of prescribed pads (Table 1), it was 0.03 (95% CI: 0.02–0.05) for PCa men before RP, which rose to 12.61 (95% CI: 11.59–13.65) at 3-month post-RP then declined to 6.71 (95% CI: 6.10–7.34) at 12 months to eventually 4.19 (95% CI: 3.76–4.62) at the end of follow-up (36 months post-RP). In contrast, this rate was 0.06 (95% CI: 0.04–0.09) and 0.12 (95% CI: 0.10–0.14) for non-PCa control men with and without BPH, respectively. Concerning post-RP phase, the monthly pad rate was 22.94 (95% CI: 20.97–24.97) for MIS RP compared to 3.16 (95% CI: 2.58–3.78) for ORP at 3 months post-RP. The considerable higher pad rate for MIS RP than that for ORP persisted at 12 months (12.19 [95% CI: 11.01–13.42] vs. 1.7 [95% CI: 1.32–2.11]) up to the 36 months follow-up (7.5 [95% CI: 6.69–8.35] vs. 1.2 [95% CI: 0.84–1.50]). Of note, the narrow confidence intervals indicated non-random differences between groups. The smoothed monthly rate of prescribed pads post-RP among patients are depicted according to RP procedure in Figure 2a.

### 3.2. Effect of Age

The effect of age on the rate of prescribed pads was significantly different between ORP and MIS procedure. The post-RP rates of prescribed pads among ORP patients were highest in men aged 45–<60 years, followed by those aged 70+ years and was lowest in those aged 60–<70 years (Figure 2b). In RP-procedure specific Poisson regressions (Appendix A), pre-RP rates of prescribed pads were significantly associated with higher post-RP pad rates in both procedures. Specific to MIS procedure, older age was significantly associated with higher rate of prescribed pad at post-RP.

### 3.3. Number of Prescribed Pads

Overall, 85% RP patients (94.6% for ORP, 75.3% for MIS RP) had no pads prescribed. Among RP patients who received any pad prescription, about 50% received a total of 1–<500 pads prescriptions per patient (Appendix A). With respect to the influence of number of operations per hospital, a significant but very weak negative association (Spearman’s r = −0.07, *p* < 0.0001) for ORP and a significant positive association (0.22, *p* < 0.0001) for MIS RP were observed.

## 4. Discussion

The present population-based longitudinal analyses utilizing insurance claims data demonstrated a significant long-term UI burden in men with PCa post-RP including all RP patients in Austria within the study period. The most interesting questions for these patients include the general risk and the intensity of post-RP UI, the time to achievement of satisfactory urinary function, and the difference between surgical procedure types (ORP vs. MIS). Our results indicate a similar preoperative pad use and significantly higher pad use after RP when compared to the general population. We also found continuous convalescence and a stable plateau in the early follow-up with only a small proportion of long-term heavy pad users.

Our results are in line with previous reports which showed continuous convalescence of UI with time [4,6,12] and only minimal changes beyond 12 months [10,13] or 24 months [17]. However, Liss et al. [18] reported that even the use of a single safety pad compared to pad free patients significantly decreases the QoL. With respect to the different RP procedures, Haglind et al. [19] reported in a prospective, controlled, non-randomized trial, comprising of 2431 RP patients from 14 centers, no significant difference in UI after MIS RP compared to ORP after 12 months. Similar outcomes were reported by Coughlin et al. [10] in a randomized controlled Phase 3 trial, comprising of 326 RP patients at 6, 12 and 24 months. In contrast, both Hu et al. [20] and Touijer et al. [21] reported higher incontinence rates after MIS RP when compared to ORP. As for a technical explanation for the difference between ORP and MIS, each one has known pros and cons, including better haptic tissue feedback for ORP and better visualization and lighting for MIS [19]. We also noted a significant higher post-operative pad rate in men who underwent MIS RP over three years. However, our findings should be considered based on their limitations. About 85% of the patients never received prescribed pads. We believe that this is not entirely based on favorable continence outcomes. Reported higher pad rate after MIS might be associated with better postoperative incontinence consulting and prescription in those centers. Additionally, due to personal preferences or administrative obstacles patients may use non-professional or professional products besides incontinence pads or purchase incontinence pads out-of-pocket by themselves. We do not see any reason for different private preference and purchasing behavior between ORP and MIS patients which might strongly bias our results, though.

Age is commonly described as a prognostic factor in literature [5,17,22], but we only observed significant age effect in pad use in the MIS RP group. Furthermore, high hospital and surgeon volume has been linked to better functional outcomes [23,24]. The volume impact, however, was observed solely in the ORP in the present analyses. Nevertheless, many hospitals show zero pad prescriptions in the follow-up. Another interesting observation was the high number of low volume centers performing under 30 RP annually. In contrast, the observed RP caseloads are much lower than internationally established high volume centers [24,25].

With therapeutic advances, localized PCa is highly treatable, thus has resulted in a growing number of patients whose survival is comparable with non-cancer men. However, many survivors live with long-lasting and distressing side effects associated with RP, which is often overlooked in clinical care. Several previous studies attempted to examine the side-effect after localized PCa treatment, mostly based on unrepresentative patient samples [3,8,19], with no inclusion of non-cancer controls, and many did not consider the type of surgery procedures. Only few studies use population-based data to measure long-term functional outcomes [4,26]. Real-world data, or administrative data, generated from insurance claims is particularly advantaged in low-cost, stretches over a long period, and contain information on the overall population in countries with universal health coverage. Furthermore, real-world data offers information of what is currently ‘achieved’ rather than what is ‘achievable’, and therefore, highlights gaps in healthcare [27]. In consideration of health care expenditure, real-world data could be a valuable tool to capture direct and indirect expenses of postoperative incontinence management, which are frequently omitted in clinical studies.

Unlike previous studies that assessed snapshot measures on self-reported UI among PCa at each time point during their study course, we were able to calculate rate measures on pad use, which provide additional information characterizing the longitudinal patterns of post-operative UI. In direct comparison to PROMs, measures of UI from administrative data are not limited by reporting bias and can accurately reflect its associated healthcare. However, pad use assessed by insurance data do not reflect the UI associated QoL. Furthermore, in comparison with other measures, e.g., the 48-h pad test, patient pad prescription count is an inferior tool to measure urinary incontinence [28]. Then again, it can identify those patients in a large collective who require additional attention. Another limitation of insurance data is the lack of information on potentially confounding factors, including disease characteristics, neurological condition, radiation history, indwelling catheter before surgery, subsequent treatment, and surgical expertise. Further studies including parameters on PCa disease characters, detailed information on the surgery procedure (i.e., nerve-sparing, preservation of the distal urethra) and QoL measures is needed to investigate the outcomes between open and MIS RP. Hence, insurance data should be, ideally, collected in a manner that it can be incorporated with well-designed research studies to easily answer unresolved research question and develop strategies in reducing UI burden in the PCa survivorship. This step would require a collaboration of a transdisciplinary team of all relevant stakeholders [29].

## 5. Conclusions

In summary, our data support the previous literature reporting a high burden of UI after RP. Although the UI symptoms generally followed high convalescence rate, ORP appeared to perform better than, or at least non-inferior to MIS RP. Future research is required to evaluate this question conclusively.

## Figures and Tables

**Figure 1 ijerph-18-06891-f001:**
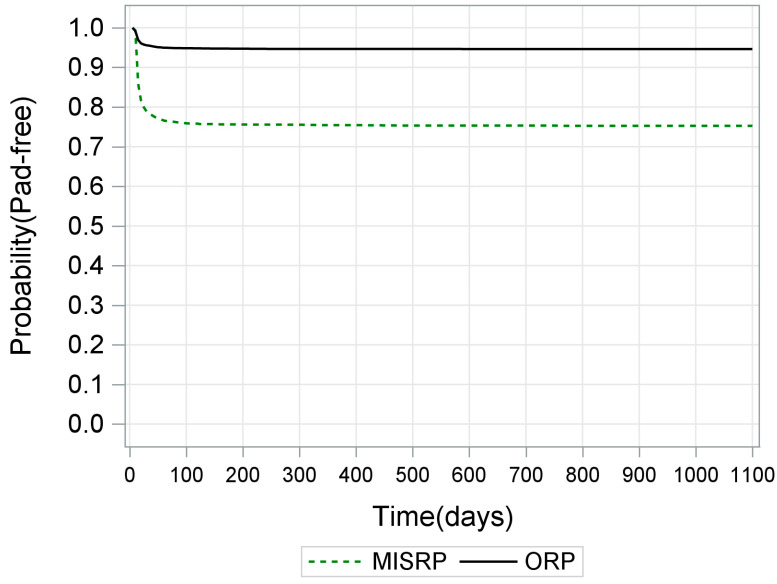
Lifetable analyses on the daily proportion of prostate cancer patients with no incontinence pad prescription (pad-free) from the surgery day through 36 months (1080 days) follow-up.

**Figure 2 ijerph-18-06891-f002:**
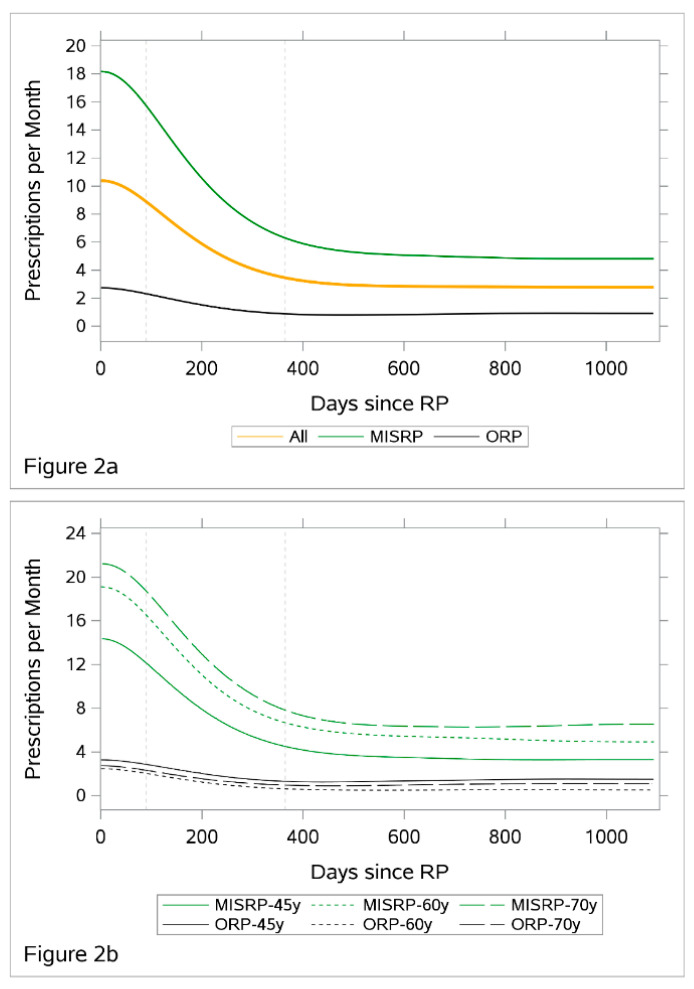
(**a**) Post-operative monthly rate of prescribed incontinence pad among prostate cancer patients overall and by RP procedures; (**b**) post-operative monthly rate of prescribed incontinence pad among prostate cancer patients overall and by RP procedures.

**Table 1 ijerph-18-06891-t001:** Sample characteristics and incontinence pad prescription among prostate cancer patients underwent radical prostatectomy between 2012–2015 in Austria and cancer-free control men.

	Prostate Cancer Patients Underwent RP ^1^			Cancer Free Controls	
	Open	Minimally Invasive	Total	With Non-Operated BPH ^2^	Without BPH
Number of Cases (*n*)	3262	2986	6248	7158	50,257
2013	1227	932	2159		
2014	1055	1008	2063		
2015	980	1046	2026		
Age (years), median (Q1–Q3)	64.6 (58.8–68.7)	64.6 (58.8–69.6)	64.6 (58.8–69.1)	70.8 (62.5–77.9)	64.3 (54.3–74.4)
Patients received incontinence pad prescription (*n*)				54	504
Preoperative	15	36	51		
Postoperative	175	737	912		
Total number of incontinence pad prescription (*n*)					
Postoperative	13,280	4971	18,251		
Total number of prescribed incontinence pad (*n*)				42,813	591,390
Preoperative (from 1/1/2011)	1613	6947	8560		
Postoperative	135,106	809,899	945,005		
Monthly rate of prescribed incontinence pads [95% CI]				0.06 (0.04–0.09)	0.12 (0.10–0.14)
Preoperative	0.01 (0.01–0.02)	0.05 (0.03–0.08)	0.03 (0.02–0.05)		
3 months postoperative	3.61 (2.58–3.78)	22.94 (20.97–24.97)	12.61 (11.59–13.65)		
12 months postoperative	1.70 (1.32–2.11)	12.19 (11.01–13.42)	6.71 (6.10–7.34)		
36 months postoperative	1.15 (0.84–1.50)	7.51 (6.69–8.35)	4.91 (3.76–4.62)		

^1^ Radical prostatectomy; ^2^ benign prostatic hyperplasia.

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
