# Peer review of "Difference in Incontinence Pad Use between Patients after Radical Prostatectomy and Cancer-Free Population with Subgroup Analysis for Open vs. Minimally Invasive Radical Prostatectomy: A Descriptive Analysis of Insurance Claims-Based Data"

_ijerph, 2021, doi:10.3390/ijerph18136891_

Round 1
Reviewer 1 Report
This is a population-based longitudinal study utilizing insurance claims data that confirms a significant long-term incontinence burden in prostate cancer men undergoing radical prostatectomy. The main strength of the study is in the use of real-world administrative data on a large study population. Most of the results are just confirmatory of what is already well known; however, it is very interesting and surprising to observe that a significantly higher post-operative pad rate was reported in men who underwent minimally invasive radical prostatectomy over three years, in comparison with those receiving open surgery.
I have some minor comments on this ms:
- The authors correctly highlighted the main limitations of their study, however, another important drawback to be mentioned is the very low accuracy of pad use as a metrics for urinary incontinence, in comparison with other objective measures and especially with pad test as demonstrated in both incontinent men and women [Sacco E, Bientinesi R, Gandi C, Di Gianfrancesco L, Pierconti F, Racioppi M, Bassi P. Patient pad count is a poor measure of urinary incontinence compared with 48-h pad test: results of a large-scale multicentre study. BJU Int. 2019 May;123(5A):E69-E78].
- Furthermore, some patients do not use pads but just homemade products for mild incontinence or condoms for severe incontinence; accordingly, some centers were associated with zero pad prescription!
- Pad prescriptions do not take into account products that patients purchase out-of-pocket that could be very relevant, depending on national healthcare systems.
- 6-line 171: “…received a total of 1- <500…” (please clarify)
- It would be very interesting to present the results also based on the type of MIS surgery: laparoscopic or robotic. The continence results may be different.
Reviewer 2 Report
Thank you for submitting interesting paper, and your results can be useful for the preoperative counselling with patients.
As your discussion, high hospital and surgeon volume has been linked to better functional outcomes. So, in order for your research results to be more useful information, further analysis is needed.
1) open vs laparoscopic vs robotic surgery should be divided.
2) more detailed analyses according to type of surgery and difference of hospital/surgeon volume are needed.
Nowadays, robotic radical prostatectomy has become main trend for surgical treatment of prostate cancer. so most urologic surgeons will not agree with your results that minimally invasive surgery shows much worse results compared to open surgery. There should be more results and comments about this point.
Round 2
Reviewer 2 Report
Thanks for your detailed revision.